# Association Between the Food Environment Around Schools and Food Consumption of Adolescents in Large and Small Municipalities in Southern Brazil

**DOI:** 10.3390/ijerph21111524

**Published:** 2024-11-16

**Authors:** Maria Beatriz Carolina da Silva, Katiany Claudete Pinheiro, Gabriele Rockenbach, Patrícia de Fragas Hinnig, Maria Gabriela Matias de Pinho, Lidiamara Dornelles de Souza, Adalberto A. S. Lopes, Francisco de Assis Guedes de Vasconcelos, Elizabeth Nappi Corrêa

**Affiliations:** 1Postgraduate Program in Nutrition, Federal University of Santa Catarina, Florianópolis 88040-370, SC, Brazil; maria.beatriz.carolina@posgrad.ufsc.br (M.B.C.d.S.); lidiamara.dornelles@sme.pmf.sc.gov.br (L.D.d.S.);; 2Department of Nutrition, Federal University of Santa Catarina, Florianópolis 88040-370, SC, Brazil; katiany.pinheiro@grad.ufsc.br; 3Center for Health Sciences, Department of Nutrition, Federal University of Santa Catarina, Florianópolis 88040-370, SC, Brazil; gabriele.r@ufsc.br; 4Faculty of Health Sciences, University of Brasília, Florianópolis 88040-370, SC, Brazil; patricia.hinnig@unb.br; 5Copernicus Institute of Sustainable Development, Department Environmental Sciences, Utrecht University, 3584 Utrecht, The Netherlands; m.g.matiasdepinho@uu.nl; 6Specialist in Geoprocessing, Physical Activity and Health, Urban Health Observatory, Federal University of Minas Gerais, Belo Horizonte 31270-901, MG, Brazil; adalberto.lopes@posgrad.ufsc.br

**Keywords:** built environment, food deserts, schools, adolescents, ultra-processed foods

## Abstract

This cross-sectional study aimed to evaluate the association between the consumption of healthy and unhealthy food markers among adolescents enrolled in the ninth grade of elementary school in municipal public schools and the food environment around the schools in two municipalities of different sizes, located in Southern Brazil. The data were collected between 2022 and 2023, with 449 adolescent participants. Of these, 347 were students from the municipality of Florianópolis, and 102 were students from the municipality of Governador Celso Ramos, all aged between 13 and 17 years. The establishments located around the schools were evaluated using AUDITNOVA, an instrument that investigates the environmental dimension and food dimension. The consumption of healthy eating markers (beans, vegetables, and fresh fruits) and unhealthy eating markers (ultra-processed foods, soft drinks, sweets, and fried snacks) among students was assessed using a food frequency questionnaire. Statistically significant associations were observed between the food environment around the school and the consumption of unhealthy food markers (OR = 0.63; 95% CI = 0.41–0.98 *p* = 0.041) but only in the large municipality. No significant associations were found in the students’ food consumption in the small municipality. A significant association between the school food environment and adolescents’ consumption of unhealthy foods was observed in Florianópolis. Healthy food consumption was low among students in the small municipality, Governador Celso Ramos.

## 1. Introduction

The food environment refers to the physical, socioeconomic, political, and cultural opportunities and conditions of a collective or individual nature that play an important role in food consumption [1]. Conceptual models conceive the food environment as a multidimensional place, which can favor the development of a food system with reduced or no availability and access to healthy foods [2,3,4].

The food environment, especially the community environment, comprises the place where individuals are inserted, such as their home, work, and school [5]. Therefore, multiple factors, such as the diversity among food retailers and the availability and access to ultra-processed foods in the school surroundings can contribute to the development of an obesogenic environment, influencing adolescents’ nutritional status [6,7]. The lack of establishments selling fresh or minimally processed foods around the school, as well as the high price of these foods, can be a challenge for students to adhere to healthy eating habits [8,9].

Schools are generally surrounded by cafeterias, grocery stores, fast food restaurants, snack bars, and convenience stores; all these establishments can determine the type of food that students have access to, reducing the supply of healthy options [10]. Strategies such as marketing actions, promotional prices, and the offer of hyperpalatable foods are evident features in these commercial food outlets in the school environment [11]. Studies indicate that the presence of certain types of businesses in the vicinity of Brazilian public schools directly influences the inappropriate consumption of unhealthy foods by students [12,13].

In Brazil, there is a regulation that provides a food service in schools for basic education students within the scope of the National School Food Program (PNAE). This emphasizes growth and biopsychosocial development, learning, academic performance, and the formation of healthy eating practices of students through food and nutritional education actions and the provision of meals that cover their nutritional needs during the school period [14]. In the state of Santa Catarina located in the south of Brazil, this municipality follows the Canteens Law (State Law nº 12.061/2001 of Santa Catarina), which establishes criteria for the concession of snacks and drinks services in the state’s educational units [15].

Scientific evidence has shown that adolescents have an inadequate diet, with high caloric density and low nutrient content. Determinants such as low adherence to school meals, demographic aspects, and the presence of cafeterias and retailers in and around the school are associated with the consumption of ultra-processed foods [16,17]. In the international and national context, the literature shows an increase in the presence of food outlets around schools, showing that physical accessibility to foods considered unhealthy constitutes a factor that favors the increase in overweight/obesity among schoolchildren [18]. In Brazil, population survey data show an increase in ultra-processed food consumption by adolescents associated with the development of chronic diseases [19,20].

Studies with adolescents conducted in Montes Claros, Minas Gerais [21], and in the city of São Paulo [22]—located in the southeast region of Brazil—identified high consumption of unhealthy foods such as ultra-processed foods, fried snacks, sweets, and soft drinks and low intake of foods such as beans, vegetables, and fresh fruits, considered healthy, also showing that socioeconomic, cultural, and environmental characteristics were crucial factors in food choices. In Barcelona, Spain [23], and in New York, United States [24], research showed that the food establishments closest to schools were fast food restaurants, convenience stores, snack bars, and food courts alongside a high prevalence of student obesity.

Although studies indicate significant inadequate nutrition among adolescents, there is still a scarcity of studies in the Brazilian literature showing disparities in the food environment between large and small cities and studies that associate the food environment in the school surroundings with the consumption of fresh, minimally processed, and ultra-processed foods by adolescents [25,26,27,28]. Thus, the present study aimed to verify the potential association between the consumption of foods that are markers of healthy and unhealthy eating habits among ninth-grade elementary school students and the food environment surrounding public schools in municipalities of different sizes located in Southern Brazil.

## 2. Materials and Methods

### 2.1. Study Design and Location

This is a cross-sectional, school-based study carried out in two municipalities with different socioeconomic and urbanization levels in the state of Santa Catarina, in the southern region of Brazil. Florianópolis, one of the municipalities assessed in this investigation, has an estimated population of 516,524 inhabitants distributed in a territory of 674.844 km^2^, with a population density of 623.68 inhabitants/km^2^ and a municipal human development index (MHDI) of 0.847, being considered a large municipality [29]. The other municipality, Governador Celso Ramos, has an estimated population of 14,739 inhabitants distributed in a territory of 127.556 km^2^, with a population density of 110.93 inhabitants/km^2^ and an MHDI of 0.747, being considered a small municipality [29,29].

### 2.2. Sample Calculation and Sample Size and Selection

To calculate the sample size, data from all municipal public schools were considered; the data were provided by the municipal education departments of each municipality. In Governador Celso Ramos, since the city is small, a census was conducted, which included all municipal public schools that had 9th-grade classes, corresponding to a total of three schools and six classes. In the municipality of Florianópolis, the sample size calculation considered the following parameters: the average observed proportion of commercial categories around Brazilian public schools [30,31] was 32.8%, with a margin of error of ±5%, and a design effect of 1.8. The estimated sample size was 509 students. Adding 30% for possible losses and 15% to enhance the sample’s capacity to assess additional variables, including food consumption, resulted in a final sample of 739 students in Florianópolis. This 15% increase ensures the representation of variability in these additional variables. The sampling process considered approximately 2 classes per school and 30 students per class. A total of 11 schools were randomly selected from 26 across the four regions of Florianópolis, and all 9th-grade students from these schools were invited to participate in this study.

### 2.3. Inclusion and Exclusion Criteria

All students whose parents and/or guardians agreed to their participation in the study and who delivered the duly signed Free and Informed Consent Form (FICF) were included. At the same time, the adolescents who agreed to participate in the survey signed and delivered the Free and Informed Assent Form (FIAF). Finally, the food establishments whose managers authorized the audit of their establishments signed the Free and Informed Consent Form (FICF) and participated in the survey. This study was conducted in accordance with the Declaration of Helsinki and the protocol was approved by the Ethics Committee of the Federal University of Santa Catarina in 2021 (Opinion Number: 4,533,681).

### 2.4. Data Collection

Data collection was performed between the years 2022 and 2023. A questionnaire (Google Forms) was applied in the schools. The form contained questions about adolescents’ food consumption, taken from the 2015 National School Health Survey [32]. The instrument also contained information on gender, age, skin color/race, mother’s education, and type of school commuting system. The adolescents completed the questionnaire individually in the school’s computer room (on computers, tablets, and cell phones), monitored by the investigators.

Subsequently, data collection was performed in the food environment around the schools in both municipalities. To this end, an 800 m network buffer was created as a coverage area [33,34] based on the street network shapefile obtained from the Open Street Map (OSM). Each school unit was considered as the centroid of the generated polygons. For the audit process—carried out on site using the AUDITNOVA [35] instrument—GPS (Global Positioning System) equipment Garmin eTrex^®^ [10] was used, with the recording of latitude and longitude to georeference the establishments that were detected selling food in the areas surveyed.

### 2.5. Instruments and Variables

#### 2.5.1. Consumption of Fresh, Minimally Processed, and Ultra-Processed Foods

Food consumption was assessed based on the frequency of consumption in the last seven days, without quantifying portion size, using a questionnaire adapted from the National School Health Survey (PeNSE), a periodic survey conducted since 2009 at the national level in Brazil. The PeNSE assesses various aspects of adolescents’ health, including eating habits, which are evaluated based on the habitual consumption of healthy food groups (beans, legumes or vegetables, and fresh fruits) and unhealthy food groups (fried snacks, sweets, soft drinks, and ultra-processed foods). Several studies have used this methodology to assess the food consumption of students in Brazil [36,37,38].

The outcome of this study was the consumption of fresh, minimally processed, and ultra-processed foods by adolescents. The analyses of data on the food consumption of 9th-grade students were described according to the frequency of consumption during the previous week (seven days).

The markers of healthy eating included the consumption of beans, vegetables, or greens and fresh fruits. The markers of unhealthy foods included the consumption of fried snacks, sweets, soft drinks, and ultra-processed foods. Based on the responses for each food separately, food consumption was categorized with the cutoff point of regular consumption (≥5×/week) and non-regular consumption (<5×/week) [39,40].

The markers were also presented as continuous scores. In this case, the frequency of consumption of each food, within its relevant groups, was added up, generating a score that ranged from 0 to 21 points for the markers of healthy eating and 0 to 28 points for the markers of unhealthy foods. The higher the score, the greater the consumption of foods from that marker, which is considered a positive behavior for healthy food markers and a negative behavior for unhealthy food markers.

For the data analysis, the scores of the healthy and unhealthy food markers were also categorized as more or less healthy, using the median as the cutoff point (median for healthy food markers: score 11.00; median for unhealthy food markers: score 10.00).

#### 2.5.2. Audit of Establishments Surrounding Schools

To construct the exposure variable for this research, an audit was carried out in each establishment that was present in the 800 m network buffer zone. The AUDITNOVA instrument was applied [35], which evaluated the food establishments’ environment dimension (advertising/promotion) and food dimension (availability and normal/promotional price). This instrument made it possible to evaluate a list of 54 foods that are classified according to the degree of processing.

In order to identify establishments as healthier or less healthy, the Consumer Food Environment Healthiness Score (CFEHS) [41] instrument was used, which assesses the healthiness of the food environment based on the Brazilian Population Food Guide [42]. This consists of two dimensions—the food dimension and the environment dimension—which standardizes a scale from 0 to 100 points, with the higher the score (closer to 100), the healthier the food sales establishment. 

In the food dimension, the total score for each indicator considered food availability and regular/promotional price and these variables were assessed dichotomously (yes/no). For example, in the case of the availability of 1 or 2 fruits, 3 points were ascribed; for 3 to 5 fruits = 6 points; for 6 fruits = 9 points. Since fruits are markers of healthy eating, 3 fruits (variable) with promotional prices scored an extra 3 points. Opposite scores were assigned to the “ultra-processed foods” indicator, considering the total count of items collected by the AUDITNOVA instrument [35], with each available ultra-processed food receiving a negative score (−1).

For the environment dimension, points were assigned according to the scoring parameter related to the Food Guide for the Brazilian Population [42], that is, if the advertising strategies in the food environment of the establishments were related to the group of natural or minimally processed foods, the score was positive, if they were related to the group of ultra-processed foods, it was negative. Considering the assessment of the establishments through the AUDITNOVA instrument [35], in order for the environment and food dimensions to have the same weight in the final score, the scores were added and then divided by two.

### 2.6. Adjustment Variables

The following adjustment variables relating to students’ individual characteristics were included: gender (male; female), age (13–14 years; 15–17 years), maternal education as a proxy for income (unknown; up to 12 years or 13 years or more schooling), reported skin color (white; non-white [black/brown/other]), type of transportation to school (active [walking/bicycle]; passive [bus/car/van]), and the municipality variable (Florianópolis; Governador Celso Ramos).

### 2.7. Statistical Analysis

Data were presented using absolute (*n*) and relative (%) frequencies for categorical variables as well as means and standard deviations (SD) for continuous variables. The normality of continuous data was confirmed by the Shapiro–Wilk test or analysis of kurtosis and skewness. The chi-square test was used to compare sample characteristics according to municipalities of residence.

The comparison of food environment scores and food consumption markers between municipalities was performed using the *t*-test for independent samples. The chi-square test was used to compare the prevalence of consumption of healthy and unhealthy food markers (equal to or greater than 5 days a week) according to the municipality of residence. Logistic regression models were calculated to analyze the association between the food environment in the school area (exposure) and healthy and unhealthy eating markers (outcomes).

Univariate analyses were performed and adjusted for confounding variables (gender, age, skin color, mother’s education, and type of school commuting system). The results are presented as Odds Ratios (ORs) with their respective 95% confidence intervals (95%CI). In these analyses, the categories of values below the median were used as reference for both the exposure and the outcome.

All data were previously planned by the research team using Microsoft Excel (version 2021) and then transferred to the statistical program IBM SPSS Statistics for Windows (Version 26.0. IBM Corp., Armonk, NY, USA) where the analyses were performed. The level of significance adopted was 95% (*p* < 0.05).

## 3. Results

Table 1 presents the students’ characteristics of each municipality of residence, Florianópolis (FLN—*n* = 347; 77.3%) and Governador Celso Ramos (GCR—*n* = 102; 22.7%). Most adolescents were female (FLN: 54.2% and GCR: 55.9%), aged between 13 and 14 years (FLN: 58.2% and GCR: 83.3%), with white skin (FLN: 58.5% and GCR: 71.6%), and used a passive commuting service to school (FLN: 56.2% and GCR: 54.9%). Regarding the mothers’ education, most of them had up to 12 years of schooling (FLN: 44.6% and GCR: 46.1%) as reported by the students. A significant difference was observed between the evaluated municipalities in terms of the age and skin color of the schoolchildren. The municipality of Governador Celso Ramos showed a higher proportion of younger, white schoolchildren compared to Florianópolis.

The spatial distribution of establishments in the schools’ neighborhoods in the cities of Florianópolis/SC (58) and Governador Celso Ramos/SC (21) are shown in Figure 1 and Figure 2. The characterization of the types of establishments that sell food in those areas and the scores for the food, environment, and general dimensions in the cities investigated are shown in Table 2. The most prevalent types of establishments in both cities were neighborhood markets (FLN: 41.4% to GCR = 47.6%) and bakeries (FLN: 17.2% to GCR = 23.8%).

The frequency of food consumption by schoolchildren and the distributions of healthy and unhealthy food markers in adolescents according to the cutoff point for regular consumption (≥5×/week) and non-regular consumption (<5×/week) can be seen in Figure 3. In general, students from both municipalities showed a higher prevalence of non-regular consumption of beans (FLN: 65.4% and GCR: 71.6%), vegetables (FLN: 49.9% and GCR: 70.6%), and fruits (FLN: 55.3% and GCR: 64.7%). Regarding markers of unhealthy food, regular consumption was prevalent for fried snacks (FLN: 8.1% and GCR: 7.8%), soft drinks (FLN: 17.9% and GCR: 14.7%), sweets (FLN: 30.8% and GCR: 29.4%), and ultra-processed foods (FLN: 31.1% and GCR: 15.7%) compared to non-regular consumption. Adolescents from Florianópolis presented higher consumption of unhealthy food markers compared to those from Governador Celso Ramos. Statistically significant differences between the two municipalities were seen for the consumption of vegetables (*p* < 0.001) and ultra-processed foods (*p* = 0.002).

The association between healthy and unhealthy food markers and the food environment score is shown in Table 3. Statistically significant associations were observed only for the city of Florianópolis, in the crude (OR = 0.63; 95%CI = 0.40–0.98; *p* = 0.040) and adjusted (OR = 0.63; 95%CI = 0.41–0.98; *p* = 0.041) analyses. In this case, students belonging to the food environment in the school surroundings with greater commercialization of ultra-processed foods (greater than the median) presented a 37% greater possibility of high consumption of unhealthy food markers. There were no significant associations for students from Governador Celso Ramos.

## 4. Discussion

This study investigated the association between the food environment around schools and the food consumption of adolescents from two cities of different sizes. In Florianópolis, which is considered a large city, an association was observed between the food environment around schools and the food consumption of unhealthy food markers among adolescents. In this environment, a high presence of establishments that sold ultra-processed foods was observed as well as many advertisements and promotional prices, facilitating the consumption of these foods among students.

On the other hand, in Governador Celso Ramos/SC, a small city, no association was observed between the food environment around schools and the food consumption of students. However, it was observed that in the surroundings of the schools evaluated in this city, there was a density of establishments that also sold predominantly unhealthy foods with a lot of advertising and promotional prices of ultra-processed foods, providing students with great availability and accessibility to this type of food. In addition, the food consumption of unhealthy food markers was shown to be relevant among students from the small city.

The expansion of food retailers in the school environment may have been one of the crucial factors in the schoolchildren’s food consumption. The most frequent establishments were neighborhood markets and bakeries. These establishments sell a variety of foods, especially ultra-processed foods [43,44].

A study conducted in Recife, in the northeast of Brazil, showed that the predominant points of sale in the vicinity of public schools sold a lot of ultra-processed foods [45]. In the city of Rio de Janeiro, Brazil, the presence of establishments selling ultra-processed foods was more frequent in the vicinity of schools, and the social segregation of food environments was associated with the low availability and accessibility of healthy foods [46].

Similar results were found in Belo Horizonte, in the state of Minas Gerais [47], in the state of Bahia [48], and in the city of Florianópolis [26], in the southeast, northeast, and south regions of Brazil, respectively, identifying a relevant number of fast-food establishments in the vicinity of public schools. In the international context, studies carried out in Ethiopia [49], California [50], and Spain [51] showed that the prevalence of overweight/obesity in adolescents was associated with the location of public schools in vulnerable areas, with low socioeconomic factors, and the school environment was seen as an obesogenic environment.

Our study did not reveal any association between the food environment in the school surroundings and the adolescents’ food consumption in Governador Celso Ramos/SC, a municipality classified as small. This result is similar to the research carried out in the Flanders region of Belgium [52], in which the school environment had a high density of establishments selling ultra-processed foods. However, it did not indicate any significant associations between the school environment and the students’ food consumption. Furthermore, a study carried out in Mexico [53] found an inverse association, showing that even with a high frequency of establishments selling unhealthy foods, adolescents presented a high consumption of healthy foods.

Another relevant factor is exposure to a greater number of retailers selling unhealthy foods on the way to school as well as advertising appeals [11,54]. Therefore, it is noteworthy that the data found in our study showed that adolescents were actively commuting to school and therefore had greater possibilities of exposure to these factors. A study carried out in New Zealand with adolescents who largely actively commuted to school showed that they were more likely to buy and consume unhealthy foods on their way as they had availability and accessibility to the sale of ultra-processed foods [55]. Similarly, in Australia, a study indicated that students who walked to school were exposed to points of sale that sold ultra-processed foods [56].

As a strong point of our study, the use of the AUDITNOVA instrument [35] stands out. This instrument proved to be suitable for auditing the food environment of establishments since it was developed for the Brazilian reality and allows for the measurement of different dimensions, investigating what the establishments sell and not just the type of establishment [57,58]. A study carried out in Governador Celso Ramos, Santa Catarina [59], evaluated the food environment in the school surroundings of three public schools using the AUDITNOVA instrument and showed that most of the audited establishments sold ultra-processed foods. The Consumer Food Environment Healthiness Score (CFEHS) [41], the overall score after applying the instrument, provides an indication of the profile of the establishments and the healthiness of the consumer’s food environment, which uses as a theoretical basis the recommendations of the NOVA classification of the Dietary Guidelines for the Brazilian Population [42].

Other strong points include the creation of the 800-meter buffer zone for considering the connectivity of the streets and having the school as the central point and not a Euclidean buffer—which only evaluates in a straight line and does not consider all the real possibilities of movement on the streets—as well as the use of on site auditing to identify the establishments in the vicinity of the schools in both municipalities. Additionally, another highlight was carrying out the survey in municipalities of different sizes since studies in the literature have evaluated the food environment in the school surroundings of only large municipalities, such as capitals and metropolitan regions, with food consumption [60,61,62].

Our investigation showed some differences between the large municipality of Florianópolis and the small municipality of Governador Celso Ramos, both located in the state of Santa Catarina in the southern region of Brazil. In the city of Florianópolis, the density of establishments, advertising, promotional prices, availability, and accessibility to a variety of types of food was higher when compared to the municipality of Governador Celso Ramos since many establishments such as supermarkets, bakeries, and convenience stores were more common in the surroundings of the schools investigated in the large municipality. On the other hand, in the small municipality of Governador Celso Ramos, the types of establishments most often present in the surroundings of the schools were neighborhood markets, bakeries, and butchers/fishmongers. Among the establishments surveyed, there were stores that offered natural and minimally processed foods such as fruits, vegetables, and fish, which are foods that constitute part of the local economy. However, we could observe that these establishments, in the small municipality, also had a high availability of ultra-processed foods. Furthermore, the number of schools investigated in the municipalities was discrepant: in Florianópolis the surroundings of eleven schools were evaluated compared to Governador Celso Ramos which had only three schools, is reflected in the number of food outlets and food accessibility in each of them.

Retailers adjust their offerings based on local demand and budgets, and there are no Brazilian regulations regarding the products sold in different types of establishments around schools. This context can result in the predominance of cheaper, less healthy products, emphasizing the need for shared responsibility. Policies to encourage the availability of healthy foods and support local suppliers are essential to mitigate the sale of ultra-processed, low-nutritional-value products.

The literature indicates an association between maternal education and dietary habits. Studies have shown that higher levels of maternal education are linked to healthier food choices, such as fresh and minimally processed foods. In contrast, lower levels of education are associated with a higher prevalence of ultra-processed food consumption within the family environment [36,37,38].

It is important to highlight that the food environment, characterized by the diversity of food retail outlets, is linked to household income and dietary habits. Affluent areas with higher income levels tend to have a higher proportion of establishments selling healthy foods, while economically vulnerable areas with lower income levels often have a higher prevalence of mixed or predominantly unhealthy food retailers, such as those selling ultra-processed foods [63,64].

The limitations of this study include the fact that the AUDITNOVA instrument does not assess establishments such as restaurants and snack bars and both can influence food consumption. Some variables that would be important in the outcome of this study were not investigated, such as the purchase of food in the vicinity of schools and the validation of the consumption of certain foods in the school environment. The absence of household income as an adjustment variable may be a limitation as it is an important determinant of purchasing power and food behavior. In this regard, we used maternal education as a proxy for income, which reflects the family’s material, intellectual, and other resources and may be linked to the acquisition, understanding, and implementation of knowledge on desirable eating behaviors [65,66]. Education can influence food choices by facilitating or restricting the ability to understand and interpret health-related information conveyed through nutritional education messages or on food labels [67].

## 5. Conclusions

It is concluded that the food environment in the school surroundings can influence students’ food consumption. Although this outcome was significant in only one of the municipalities investigated, adolescents in both municipalities presented a high consumption of ultra-processed foods. Adolescents belonging to the food environment in the school surroundings that showed greater sales of ultra-processed foods (higher than the median) presented a 37% greater possibility of high consumption of markers of unhealthy eating. It is necessary to continue studies in municipalities with different realities, in order to investigate the impact of the food environment on food consumption and health in the population.

## Figures and Tables

**Figure 1 ijerph-21-01524-f001:**
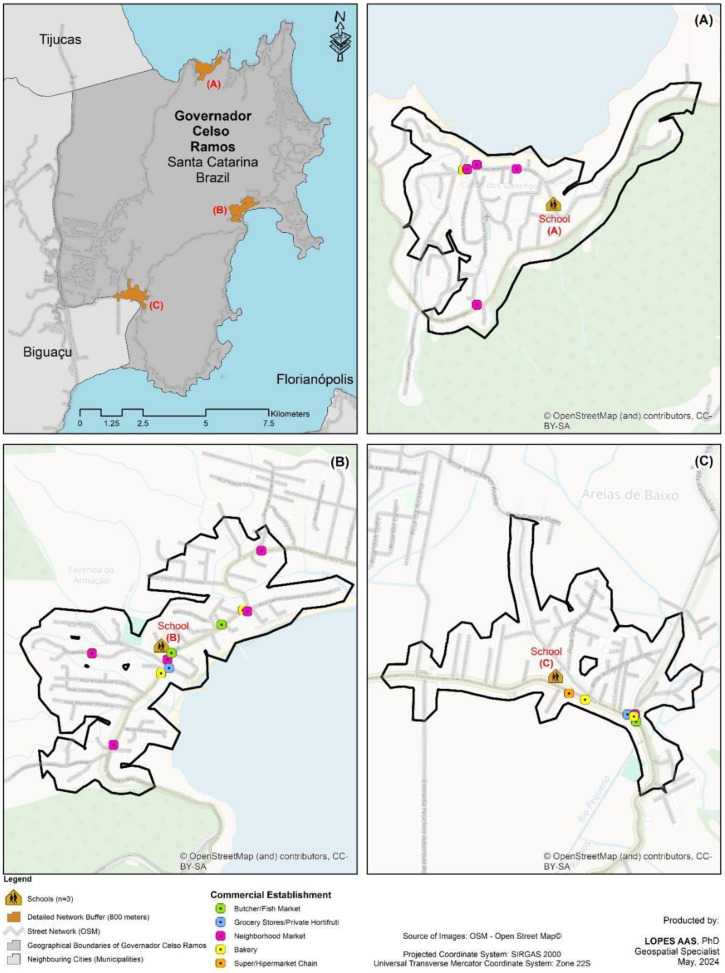
Spatial distribution of food establishments around schools in the municipality of Governador Celso Ramos/SC.

**Figure 2 ijerph-21-01524-f002:**
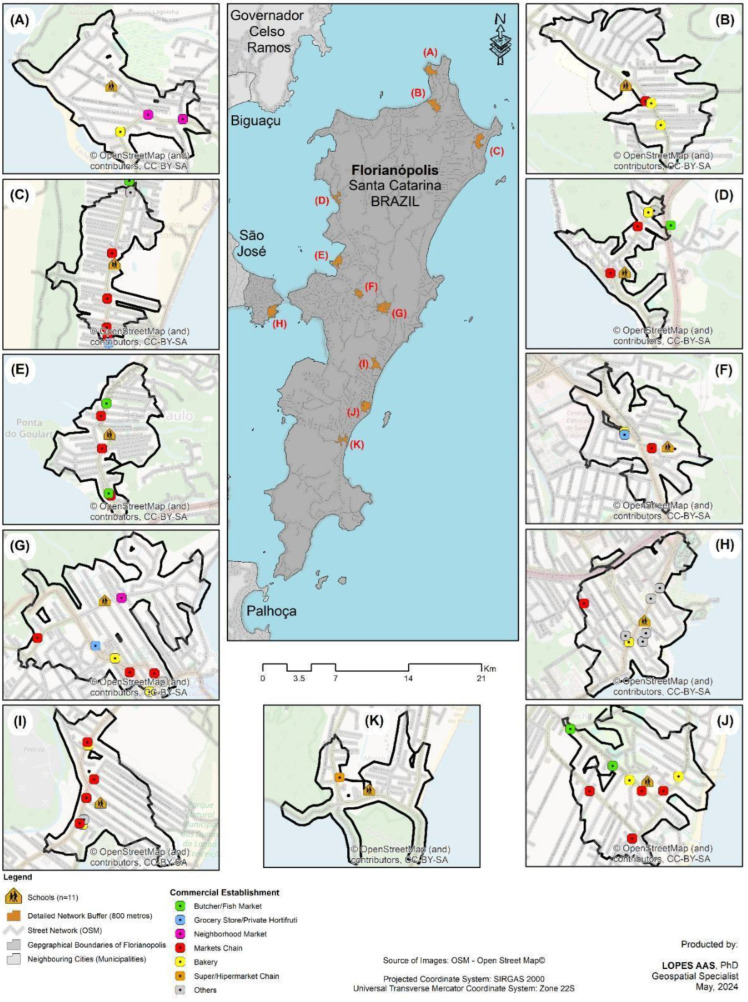
Spatial distribution of food establishments around schools in the municipality of Florianópolis/SC.

**Figure 3 ijerph-21-01524-f003:**
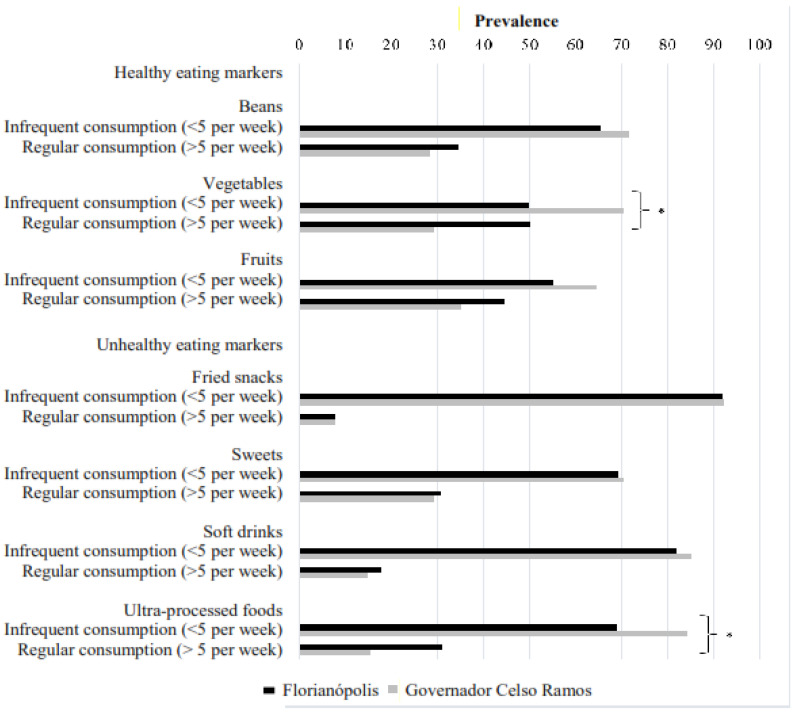
Prevalence of healthy and unhealthy food markers’ consumption among adolescents according to the municipality of residence. * Statistically significant comparison between municipalities (*p* < 0.05). Chi-square test. Source: authors.

**Table 1 ijerph-21-01524-t001:** Characterization of adolescents according to municipality of residence 2022–2023.

	Florianópolis		Governador Celso Ramos		*p* *
	*n*	%	N	%	
Total	347	100.0	102	100.0	
**Gender**					
Female	188	54.2	57	55.9	0.761
Male	159	45.8	45	44.1	
**Age range**					
13–14 years	202	58.2	85	83.3	<0.001
15–17 years	145	41.8	17	16.7	
**Skin color**					
White	203	58.5	73	71.6	0.017
Non-white	144	41.5	29	28.4	
Brown	96	27.7	18	17.6	
Black	34	9.8	8	7.8	
Indigenous	9	2.6	1	1.0	
Yellow	5	1.4	2	2.0	
**Commute**					
Active	152	43.8	46	45.1	0.817
Bicycle	20	5.8	3	2.9	
Walking	132	38.0	43	42.2	
Passive	195	56.2	56	54.9	
Car or Van	65	18.7	6	5.9	
Bus	112	32.3	44	43.1	
Other	18	5.2	6	5.9	
**Mother’s education**					
Does not know	96	27.7	26	25.5	0.910
Up to 12 years of schooling	155	44.6	47	46.1	
13 years or more of schooling	96	27.7	29	28.4	

* Pearson’s chi-square test.

**Table 2 ijerph-21-01524-t002:** Characterization of the types of food-selling establishments in the municipalities of Governador Celso Ramos and Florianópolis.

Food Retailers	Total		Food Dimension Score		Environment Dimension Score		Overall Score	
	*n*	%	Mean	SD	Mean	SD	Mean	SD
**Florianópolis**								
Neighborhood markets	24	41.4	51.1	11.6	45.1	17.3	48.1	11.1
Bakeries	10	17.2	34.3	6.2	51.8	15.5	43.1	7.0
Others ^a^	8	13.8	23.9	7.2	28.4	18.5	26.2	12.5
Butchers/Fish markets	6	10.3	42.8	5.3	57.6	16.9	50.2	10.1
Municipal fruit and vegetable markets	4	6.9	61.7	16.2	61.4	15.5	61.6	14.5
Supermarkets	3	5.2	61.0	11.2	51.5	29.2	56.3	20.1
Municipal fruit and vegetable markets	2	3.4	66.3	15.3	59.1	19.3	62.7	17.3
Hypermarkets	1	1.7	60.2	.	45.5	.	52.8	.
**TOTAL**	58	100%	45.5	15.6	47.2	19.0	46.4	14.5
**Governador Celso Ramos**								
Neighborhood markets	10	47.6	49.9	8.5	46.4	13.2	48.1	8.3
Bakeries	5	23.8	28.7	5.2	34.5	7.6	31.6	5.8
Butchers/Fish markets	3	14.3	42.6	3.0	54.5	0.0	48.6	1.5
Private fruit and vegetable markets	2	9.5	69.9	5.1	72.7	0.0	71.3	2.6
Supermarkets	1	4.8	50.6	.	36.4	.	43.5	.
**TOTAL**	21	100%	45.7	13.5	46.8	14.5	46.2	12.5

^a^ Category comprising seven gas station convenience stores and one emporium. Source: authors.

**Table 3 ijerph-21-01524-t003:** Association between healthy and unhealthy food markers and the food microenvironment score by municipality.

	Healthy Food Markers		Unhealthy Food Markers	
	OR (95%CI)	*p*-Value	OR (95%CI)	*p*-Value
**FLORIANÓPOLIS**				
**Gross**				
Food Score	1.24 (0.81–1.89)	0.322	0.94 (0.61–1.43)	0.760
Environment Score	1.24 (0.80–1.90)	0.333	0.75 (0.49–1.15)	0.185
Total Score	1.15 (0.74–1.78)	0.537	0.63 (0.41–0.98)	**0.040**
**Adjusted ^a^**				
Food Score	1.19 (0.77–1.83)	0.439	0.89 (0.58–1.37)	0.602
Environment Score	1.28 (0.83–1.99)	0.267	0.77 (0.50–1.19)	0.236
Total Score	1.13 (0.73–1.77)	0.580	0.63 (0.40–0.98)	**0.041**
**GOVERNADOR CELSO RAMOS**				
**Gross**				
Food Score	0.77 (0.33–1.82)	0.556	1.43 (0.62–3.26)	0.402
Environment Score	1.06 (0.45–2.47)	0.895	1.53 (0.68–3.45)	0.304
Total Score	1.06 (0.45–2.47)	0.895	1.53 (0.68–3.45)	0.304
**Adjusted ^a^**				
Food Score	0.66 (0.24–1.76)	0.401	0.98 (0.37–2.56)	0.966
Environment Score	1.12 (0.44–2.81)	0.817	1.30 (0.53–3.22)	0.565
Total Score	1.12 (0.44–2.81)	0.817	1.30 (0.53–3.22)	0.565

^a^ Adjusted for gender, age group, skin color, mother’s education, and commute type. For students’ dietary markers and food microenvironment scores, values below the median were used as reference. Source: authors.

## Data Availability

Data is unavailable due to privacy or ethical restrictions.

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
