# Peer review of "Association Between the Food Environment Around Schools and Food Consumption of Adolescents in Large and Small Municipalities in Southern Brazil"

_ijerph, 2024, doi:10.3390/ijerph21111524_

Round 1

Reviewer 1 Report

Comments and Suggestions for Authors

Dear Authors,

The following comments are intended to strengthen the manuscript and increase its readability:

Abstract: The conclusion replicates the result regarding the food environment and the consumption of "unhealthy" food among students in a large municipality. Since the authors also mentioned "healthy" food in the study's aims, it would be necessary to address this issue as well. Were any relationships observed concerning the consumption of "healthy" food? It should also be stated how many participants were in a small municipality and how many were in a large one - not just the total number of all respondents.

Introduction: General remark: The authors use the general terms "unhealthy" and "healthy" food - however, it is worth specifying what types of products they have included in these two groups. At the same time, it is worth determining what kinds of "unhealthy" food (fast food/ultra-processed food?) are most often consumed by teenagers in Brazil. At this point, it is also worth referring to school meals: are there any legal regulations regarding products/meals served in school canteens or school shops?

Materials and methods: Sample calculation: Please explain more on “15% for the evaluation of other variables that include food consumption” as, at this point, it is not clear. There is no information on how many respondents were planned to be included from each municipality. There was only a total number given.

Line 137: The types of establishments should be clarified. Were only McDonald' s-type restaurants (fries, hamburgers, hot dogs) considered fast food, or were they others? What other types?

Was only the frequency of consumption assessed per day (yes/no) or were the daily frequency and portion size also assessed? This should be clarified. On what basis were the markers of healthy food adopted? I wonder about the lack of e.g. fish, whole-grain products or dairy products - especially fermented ones and water as a drink. In line 136, the authors mention places selling fast food but later describe a tool they use to classify places as "healthier" or less "healthy" - this is not entirely clear to the reader. Please provide an additional explanation to clear it up.

It seems that the characteristics of respondents lack a question about economic status. This appears to be an important factor in purchasing behavior. Please comment.

Results: I suggest doing a statistical analysis on the similarities/differences between the two groups. It seems that, for example, in GCR, younger respondents dominated. This could have influenced their eating and shopping behavior.

Page 10 is blank?

Discussion: This part is well-run in my opinion. However, the limitation should include the lack of information about financial status - this is a very important factor in dietary behavior.

General remark: It seems that the authors did not assess the availability of products in the schools themselves? Did they observe any differences? This may have an impact on purchasing food outside of school.

Author Response

Dear reviewer,

The authors are grateful for the precious time spent reviewing and commenting on each relevant issue in order to improve this manuscript. Each note directed by the Editor and Reviewers was duly answered and justified, point-by-point, indicating the page and the line where it is possible to check eventual modifications, insertions and/or corrections that were made. Furthermore, a marked-up copy of the manuscript that highlights changes made to the original version is uploaded and submitted together.

- Abstract: The conclusion replicates the result regarding the food environment and the consumption of "unhealthy" food among students in a large municipality. Since the authors also mentioned "healthy" food in the study's aims, it would be necessary to address this issue as well. Were any relationships observed concerning the consumption of "healthy" food? It should also be stated how many participants were in a small municipality and how many were in a large one - not just the total number of all respondents.

Response: We appreciate the contributions made to improve the manuscript. The reviewer's suggestions have been incorporated into the new version (page 1, line 28-30, line 32-34, line 37-39).

- Introduction: General remark: The authors use the general terms "unhealthy" and "healthy" food - however, it is worth specifying what types of products they have included in these two groups. At the same time, it is worth determining what kinds of "unhealthy" food (fast food/ultra-processed food?) are most often consumed by teenagers in Brazil. At this point, it is also worth referring to school meals: are there any legal regulations regarding products/meals served in school canteens or school shops?

Response: Once again, we would like to thank you for your careful review and insightful questions. In the revised version of the manuscript, we have endeavored to incorporate the information requested by the reviewer into the Introduction section.

- In the present manuscript, we have considered "healthy foods": unprocessed or minimally processed foods (beans, legumes, vegetables, and fresh fruits). And as "unhealthy foods": ultra-processed foods (fried snacks, sweets, soft drinks, and ultra-processed foods). (page 2, line 73-74).

“In Brazil, there is a regulation that provides for a food service school for basic education students within the scope of the National School Food Program (PNAE). Emphasizing, growth and development biopsychosocial, learning, academic performance and the formation of healthy eating practices of students, through food and nutritional education actions and the provision of meals that cover their nutritional needs during the school period. In the state of Santa Catarina, located in the south of Brazil, the minicipios follow the Canteens Law (State Law nº 12.061/2001 of Santa Catarina), which establishes criteria for the concession of snacks and drinks services in the state's educational units.” (page 2,  line 58-64).

- Materials and methods: Sample calculation: Please explain more on “15% for the evaluation of other variables that include food consumption” as, at this point, it is not clear. There is no information on how many respondents were planned to be included from each municipality. There was only a total number given.

Response: We have rewritten the sample calculation to give more detail on this process. (Page 2, line 99-100 and page 3 line 101-106).

- Line 137: The types of establishments should be clarified. Were only McDonald' s-type restaurants (fries, hamburgers, hot dogs) considered fast food, or were they others? What other types?

Response: Thank you for your observation. We acknowledge that the previous wording was indeed inadequate and failed to convey the necessary information. The issue has been addressed on page 3, line 126, by changing "fast food" to "food".

Was only the frequency of consumption assessed per day (yes/no) or were the daily frequency and portion size also assessed? This should be clarified. On what basis were the markers of healthy food adopted? I wonder about the lack of e.g. fish, whole-grain products or dairy products - especially fermented ones and water as a drink.

Response: Thank you for the opportunity to clarify the question. 

“Food consumption was assessed based on the frequency of consumption in the last seven days, without quantifying portion size, using a questionnaire adapted from the National School Health Survey (PeNSE), a periodic survey conducted since 2009 at the national level in Brazil. The PeNSE assesses various aspects of adolescents' health, including eating habits, which are evaluated based on the habitual consumption of healthy food groups (beans, legumes or vegetables, and fresh fruits) and unhealthy food groups (fried snacks, sweets, soft drinks, and ultra-processed foods). Several studies have used this methodology to assess the food consumption of students in Brazil.” (page 3,  line 129-135). 

- In line 136, the authors mention places selling fast food but later describe a tool they use to classify places as "healthier" or less "healthy" - this is not entirely clear to the reader. Please provide an additional explanation to clear it up.

Response: Thank you for your observation. We would like to inform you that, in fact, the wording was inadequate and did not accurately convey the necessary information, which may have led to confusion on the part of the reader. The adjustment has been made on page 3, line 126, changing "fast food" to "food."

Additionally, we would like to inform you that further explanations have been included in the manuscript (page 4, line 157-162).

“In order to identify establishments as healthier or less healthy, the Consumer Food Environment Healthiness Score (CFEHS),43 instrument was used, which assesses the healthiness of the food environment based on the Brazilian Population Food Guide,44 consisting of two dimensions called the food dimension and the environment dimension, which standardizes a scale from 0 to 100 points, with the higher the score (closer to 100), the healthier the food sales establishments are considered for the assessment. general healthiness of the food environment.”

- It seems that the characteristics of respondents lack a question about economic status. This appears to be an important factor in purchasing behavior. Please comment.

Response: Thank you very much for your relevant and pertinent question. Indeed, in the present study, the variable economic status (family income) was not analyzed, which reveals a methodological limitation, as it constitutes an important determinant of the purchasing power/behavior of food. We clarify that in this study, we considered the mother’s education level as a proxy variable for family income or economic status. On the other hand, we clarify that the literature shows that the lower the level of education of the student's family, the higher the consumption of ultra-processed foods. And, if the level of education is equal to or greater than 12 years of study, there is an improvement in the consumption of in natura and minimally processed foods. We also highlight that we have made changes to the manuscript explaining more clearly the use of the mother’s education level as a proxy for income and its influence on purchasing power and eating behavior (page 12, line 382-385). In addition, we have included as a limitation of the study the lack of collection of information on economic status (page 13, line 394-400).

- Results: I suggest doing a statistical analysis on the similarities/differences between the two groups. It seems that, for example, in GCR, younger respondents dominated. This could have influenced their eating and shopping behavior.

Response: Thanks for the suggestion. We run a chi-squared test to check for differences between the characteristics of the schoolchildren and the municipality (Table 1). We have included a description of the test in the Methods section and described the results in the Results section (page 6, line 228)

- Page 10 is blank?

Response: We appreciate your observation and would like to inform you that there may have been an error with the PDF file downloaded from the journal system. Indeed, we have reviewed the submitted manuscript file and observed that the page is filled with Table 3 and the beginning of the discussion.

- Discussion: This part is well-run in my opinion. However, the limitation should include the lack of information about financial status - this is a very important factor in dietary behavior.

Response: Thank you again for your helpful feedback. We've added a limitation to the manuscript noting the absence of analysis on financial status, as you suggested. (page 13, line 394-400)

- General remark: It seems that the authors did not assess the availability of products in the schools themselves? Did they observe any differences? This may have an impact on purchasing food outside of school.

Response: Thank you for your question. We would like to clarify that this study focused solely on dimensions of the food environment surrounding schools. It was not our aim to evaluate dimensions of the school food environment (inside schools). We did not assess the presence of school canteens, the sale of food in schools, foods brought from home by schoolchildren, or foods provided by the National School Feeding Program (PNAE). Indeed, all of these dimensions may have an impact on purchasing food outside of school and on the reported food consumption, but such information was not collected. The differences were observed according to each municipality. However, we clarify that in both Florianópolis and Governador Celso Ramos, schools did not have canteens with the sale of food inside the schools, and in both municipalities the PNAE is regulated ( Brasil, 2020; Brasil, 2001).

Reviewer 2 Report

Comments and Suggestions for Authors

-Each age group has its own consumption habits. Therefore, as the age increases and the level of education increases, there will be a change in consumption habits. Therefore, why was this survey study not conducted on a certain number of consumers of a certain age, but in a cluster form?

-The capacity of food business operators around small municipalities and the purchasing power of the consumer are the main factors determining the degree of food quality. This is a fact. Therefore, business operators sell products to the consumer's budget. This is one of the main factors that threaten health. Therefore, how do they allow the sale of such products regardless of the size and size of the municipalities?

- As can be seen from the table, the amount of products (food size) purchased from large markets is small, while it is more than businesses around small municipalities. Which environmental factors and food dimensions are more effective and prominent here?

-Again, what are the levels of family education of people in this age group? This is also the most important factor affecting shopping and food size.

-It has not been fully stated to what extent the male and female characteristics of people shopping from these businesses are effective?

- One of the most important issues is the annual income level of the family, which is one of the most important issues that determines which of these types of food businesses consumers will be more likely to go to. These should also be included in the text.

- Consumers mostly consume fried snacks. I wonder how often the frying oils here are refreshed daily. Because oils can turn into carcinogenic substances after a period of high temperature and long-term processing. This is the most important issue that threatens human health. Comments should be made about this section.

Comments on the Quality of English Language

should be improved

Author Response

Dear reviewer,

The authors are grateful for the precious time spent reviewing and commenting on each relevant issue in order to improve this manuscript. Each note directed by the Editor and Reviewers was duly answered and justified, point-by-point, indicating the page and the line where it is possible to check eventual modifications, insertions and/or corrections that were made. Furthermore, a marked-up copy of the manuscript that highlights changes made to the original version is uploaded and submitted together.

- Each age group has its own consumption habits. Therefore, as the age increases and the level of education increases, there will be a change in consumption habits. Therefore, why was this survey study not conducted on a certain number of consumers of a certain age, but in a cluster form?

Response: We clarify that, regarding the population or age group investigated, the design of this study was based on the National School Health Survey (PeNSE) (Brazil, 2016) design. Thus, in each sampled school, classes of students enrolled in the 9th grade of elementary school were investigated (clusters), reducing the variability of age and educational level among students. Typically, 9th-grade students in Brazil are 13-14 years old, although there may be older students, albeit in smaller proportions. In these 9th-grade classes, students' ages ranged from 13 to 17 years. Therefore, age was considered as a covariate in the logistic regression analysis.

- The capacity of food business operators around small municipalities and the purchasing power of the consumer are the main factors determining the degree of food quality. This is a fact. Therefore, business operators sell products to the consumer's budget. This is one of the main factors that threaten health. Therefore, how do they allow the sale of such products regardless of the size and size of the municipalities?

Response: We appreciate your observation and would like to inform you that the discussion has been expanded (page 12, line 377-381). 

"Retailers adjust their offerings based on local demand and budgets, and there are no Brazilian regulations regarding the products sold in different types of establishments around schools. This context can result in the predominance of cheaper, less healthy products, emphasizing the need for shared responsibility. Policies to encourage the availability of healthy foods and support local suppliers are essential to mitigate the sale of ultra-processed, low-nutritional value products."

- As can be seen from the table, the amount of products (food size) purchased from large markets is small, while it is more than businesses around small municipalities. Which environmental factors and food dimensions are more effective and prominent here?

Response: We appreciate the opportunity to clarify this point. The data presented in Table 2 refers to the quantity of establishments available in the vicinity of the schools studied, and it is not possible to identify which establishments are most frequently used by students.

- Again, what are the levels of family education of people in this age group? This is also the most important factor affecting shopping and food size.

Response: Maternal education levels are presented in Table 1. as described in the Results: “Regarding the mothers’ education, most of them had up to 12 years schooling (FLN: 44.6% and GCR: 46.1%) as reported by the students.”. We have explored the relationship between mothers' education and food access and consumption in our discussion (page 12, line 382-385).

- It has not been fully stated to what extent the male and female characteristics of people shopping from these businesses are effective?

Response: Thank you for your observation. We would like to clarify that the study aimed to verify the potential association between the consumption of foods that are markers of healthy and unhealthy eating habits among 9th-grade elementary school students and the food environment surrounding public schools in municipalities of different sizes located in southern Brazil. It is not possible to identify the purchasing profile of the interviewees. The variables of sex (male and female) were used as covariates to assess the association between XX and XY. Adjustment variables, including sex, are described in the footnote of Table 3. No association was observed between sex and the total, environment, and food scores (p > 0.05) (data not shown in tables).

- One of the most important issues is the annual income level of the family, which is one of the most important issues that determines which of these types of food businesses consumers will be more likely to go to. These should also be included in the text.

Response: We thank you for your important observation and suggestion for inclusion in the manuscript. Indeed, there is consistent evidence that family income influences the type of commercial establishment where families purchase their food. Those with higher family incomes tend to shop at supermarkets, while those with lower incomes tend to purchase food at small markets or grocery stores. We have accepted the reviewer's suggestion and included comments on this matter in the discussion section (page 12, line 386-390). Additionally, we have included as a limitation of the study the fact that we did not collect information on family income (page 13, line 394-400).

- Consumers mostly consume fried snacks. I wonder how often the frying oils here are refreshed daily. Because oils can turn into carcinogenic substances after a period of high temperature and long-term processing. This is the most important issue that threatens human health. Comments should be made about this section.

Response: We clarify that, according to the data presented in Figure 3, the frequency of irregular consumption (< 5 times per week) of fried snacks among the adolescents investigated was 8.1% in the larger municipality and 7.8% in the smaller municipality (Figure 3). Indeed, the consumption of fried snacks can lead to serious health consequences. Unfortunately, the study does not have or collected information on the frequency of frying oil renewal. We thank the reviewer for his observation and clarify that the information on the consumption of ultra-processed foods is included in the manuscript (page 2, line 70-71).

Round 2

Reviewer 1 Report

Comments and Suggestions for Authors

Thank to the Authors for all explanations and amends.